# Linear and Machine Learning modelling for spatiotemporal disease predictions: Force-of-Infection of Chagas disease

Julia Ledien[1]*, Zulma M. Cucunubá[2,3], Gabriel Parra-Henao[4,5], Eliana Rodríguez-Monguí[6], Andrew P. Dobson[7], Susana B. Adamo[8], María-Gloria Basáñez[2], Pierre Nouvellet[1]

1 School of Life Sciences, University of Sussex, Falmer, Brighton, United Kingdom, 2 London Centre for Neglected Tropical Disease Research & MRC Centre for Global Infectious Disease Analysis, School of Public Health, Imperial College London, London, United Kingdom, 3 Departamento de Epidemiología Clínica y Bioestadística, Facultad de Medicina, Universidad Pontificia Javeriana, Bogotá, Colombia, 4 Centro de Investigación en Salud para el Trópico, Universidad Cooperativa de Colombia, Santa Marta, Colombia, 5 National Institute of Health, Bogotá, Colombia, 6 Independent consultant to the Neglected, Tropical and Vector Borne Diseases Program, Pan American Health Organization (PAHO), Bogota, Colombia, 7 Department of Ecology and Evolutionary Biology, Princeton University, Princeton, New Jersey, United States of America, 8 Center for International Earth Science Information Network (CIESIN), Columbia Climate School, Columbia University, New York, New York, United States of America

* j.ledien@sussex.ac.uk

**Data Availability Statement:** The datasets used for these analyses are available in the GitHub repository for Chagas disease FoI with Linear

## Abstract

### Background

Chagas disease is a long-lasting disease with a prolonged asymptomatic period. Cumulative indices of infection such as prevalence do not shed light on the current epidemiological situation, as they integrate infection over long periods. Instead, metrics such as the Force-of-Infection (FoI) provide information about the rate at which susceptible people become infected and permit sharper inference about temporal changes in infection rates. FoI is estimated by fitting (catalytic) models to available age-stratified serological (ground-truth) data. Predictive FoI modelling frameworks are then used to understand spatial and temporal trends indicative of heterogeneity in transmission and changes effected by control interventions. Ideally, these frameworks should be able to propagate uncertainty and handle spatiotemporal issues.

### Methodology/principal findings

We compare three methods in their ability to propagate uncertainty and provide reliable estimates of FoI for Chagas disease in Colombia as a case study: two Machine Learning (ML) methods (Boosted Regression Trees (BRT) and Random Forest (RF)), and a Linear Model (LM) framework that we had developed previously. Our analyses show consistent results between the three modelling methods under scrutiny. The predictors (explanatory variables) selected, as well as the location of the most uncertain FoI values, were coherent across frameworks. RF was faster than BRT and LM, and provided estimates with fewer extreme

Models. Available from: https://github.com/jledien/Chagas-disease-FoI-with-Linear-Models.git.

**Funding:** J.L. was funded by the award of a School of Life Sciences, University of Sussex PhD Studentship (to P.N.). M.G.B. acknowledges funding from the Medical Research Council (MRC) Centre for Global Infectious Disease Analysis (MR/R015600/1), jointly funded by the UK MRC and the UK Foreign, Commonwealth & Development Office (FCDO), under the MRC/FCDO Concordat agreement and is also part of the European and Developing Countries Clinical Trials Partnership (EDCTP2) programme supported by the European Union. The funders had no role in study design, data collection and analysis, decision to publish, or preparation of the manuscript.

**Competing interests:** The authors have declared that no competing interests exist.

values when extrapolating to areas where no ground-truth data were available. However, BRT and RF were less efficient at propagating uncertainty.

## Conclusions/significance

The choice of FoI predictive models will depend on the objectives of the analysis. ML methods will help characterise the mean behaviour of the estimates, while LM will provide insight into the uncertainty surrounding such estimates. Our approach can be extended to the modelling of FoI patterns in other Chagas disease-endemic countries and to other infectious diseases for which serosurveys are regularly conducted for surveillance.

### Author summary

Metrics such as the per susceptible rate of infection acquisition (Force-of-Infection) are crucial to understand the current epidemiological situation and the impact of control interventions for long-lasting diseases in which the infection event might have occurred many years previously, such as Chagas disease. FoI values are estimated from serological age profiles, often obtained in a few locations. However, when using predictive models to estimate the FoI over time and space (including areas where serosurveys had not been conducted), methods able to handle and propagate uncertainty must be implemented; otherwise, overconfident predictions may be obtained. Although Machine Learning (ML) methods are powerful tools, they may not be able to entirely handle this challenge. Therefore, the use of ML must be considered in relation to the aims of the analyses. ML will be more relevant to characterise the central trends of the estimates while Linear Models will help identify areas where further serosurveys should be conducted to improve the reliability of the predictions. Our approaches can be used to generate FoI predictions in other Chagas disease-endemic countries as well as in other diseases for which serological surveillance data are collected.

## Introduction

Chagas disease is a neglected tropical disease estimated to affect between 6 and 7 million persons worldwide. While only endemic in 21 countries in Latin America, the number of Chagas disease cases detected in Europe, North America, and the Far East has greatly increased, due to migration of infected populations [1]. Being able to identify how the cases are distributed in space and whether the control interventions implemented have been successful is critical to identifying how resources should be allocated to eliminate the disease as a public health problem in the 2021–2030 time horizon [2]. As a long-lasting and chronic disease, analyses based solely on the current prevalence of infection (typically measured as seroprevalence) has limited scope. Indeed, the prevalence recorded at a given time does not reflect the current epidemiological situation, as infection may have occurred in the past. The Force-of-Infection (FoI), i.e. the rate at which susceptible individuals become infected, is a modelling-derived metric that can be used to understand changes in incidence in space and time as a result of deliberate control interventions and/or secular changes, including environmental change [3]. However, the use of FoI raises its own challenges, particularly those regarding quantification and propagation of uncertainty when used as a response variable in predictive models. A catalytic model

(fitted to age-structured seroprevalence data, often using Bayesian methods) has been used to obtain the FoI and thus, the FoI values for each serosurvey and each year are posterior distributions and not only single values [4]. When the derived FoI is used to fit predictive models, the mean or median values of FoI are predominantly used, neglecting the uncertainty surrounding the estimated values [5–7]. Furthermore, when a non-constant (e.g. a yearly-varying) FoI is assumed, each serosurvey analysed becomes a temporal series at a certain location, requiring specific and computationally-intensive methods to be included into predictive models [8]. Machine Learning methods could represent a faster and more flexible framework to implement such models.

Machine Learning (ML) methods are computational processes based on probabilities and algorithms that use prior knowledge to produce predictions. ML can handle non-linear and non-parametric models that are able to flout the linearity, normality (Gaussian distribution) and equal variance assumptions of statistical models [9]. Essentially, ML methods make no assumptions about the statistical distribution of the data [9].

These methods have previously been used in contexts in which those assumptions are challenged, such as spatial, temporal and spatiotemporal analyses of infectious diseases, e.g. mapping of human leptospirosis [10,11], severe fever with thrombocytopenia syndrome [12], lymphatic filariasis [13], or to identify individuals with a higher risk of HIV infection based on socio-behavioural-driven data [14].

Two types of ML models have been extensively used in the context of infectious disease epidemiology, namely, Boosted Regression Trees (BRT) and Random Forest (RF). Although they are not spatial approaches (as data locations and sampling patterns are ignored to produce estimates), they have shown potential in spatial modelling [15], in particular, when used with appropriate sampling strategies [16]. Specifically, BRT and RF have been used to study the spatial spread of numerous infectious diseases, including epidemics among swine farms in the USA [17], Ebola case-fatality ratio [18], risk factors for visceral leishmaniasis [19,20], African swine fever [21], scrub typhus [22], dengue incidence [23], and dengue FoI [5]. RF also proved its potential in modelling epidemics in a spatiotemporal framework, outperforming time series models [17].

This paper aims to compare the performance of two ML methods, namely, BRT and RF, with a Linear Model (LM) framework we have previously developed [8] in their ability to predict the FoI of Chagas disease across space and time. We use detailed data from Colombia as a case study and describe the advantages and disadvantages of using such Machine Learning methods compared to Linear Model frameworks, specifically focussing on their ability to handle uncertainty on the response variable.

## Methods

### 1. Data sources

Current and past exposure to Chagas disease can be characterised by estimating the (time-varying) Force-of-Infection (FoI), i.e. temporal changes in the per susceptible rate of parasite acquisition [3,4]. Using results of 76 age-stratified serosurveys conducted at municipal level in Colombia between 1998 and 2014 (S1 Fig), yearly-varying FoI values were estimated, for each serosurvey, by fitting a catalytic model to age-stratified seropositivity data (see [4] for details). For each serosurvey, FoI estimates, for the period ranging from the birth of the oldest participants to the year when the serosurvey was conducted, were obtained using a Bayesian framework to fit the catalytic model to data, thus allowing for extraction of the full joint posterior distribution of the yearly FoI estimates. We refer to those municipalities where at least one serosurvey was conducted as municipalities 'in catchment areas', whereas those municipalities for

which serosurveys were not conducted, not available, or not used in our analyses, are referred as 'out of catchment areas'. S1 Fig in the Supporting Information file depicts the geographical distribution of the available serosurveys ('ground-truth' data).

The predictors used in these analyses included demographic, entomological and climatic variables (recorded at the municipality level), contextual information about the serosurveys (location, year conducted and setting, i.e. urban, rural, mixed and indigenous (as defined in [4]), and information from public blood banks on the prevalence of Chagas disease and number of blood units tested (available at the departmental level). A full list and description of the predictors is available in S1 Table of the Supporting Information File.

## 2. Linear Model (LM) framework

The LM framework relied on a list of plausible linear combinations of predictors selected based on expert knowledge and to avoid correlation between predictors. They were then integrated into an ensemble model using model averaging with weights based on the performance indicators of each individual linear model. The 10 best models for each setting type (urban, rural and indigenous) were averaged and used to obtain FoI predictions. The LM framework has been fully described in [8].

## 3. Machine Learning (ML) framework

Both ML methods tested in this paper are based on decision trees. A decision tree is an intuitive process that builds an algorithm by generating a step-by-step tree, whereby the dataset is repeatedly split to make a decision at each node. The splitting relies on optimising a variable-specific threshold that best discriminates the data into two branches at each node. Sequentially, the entire dataset is divided by defining new variable-specific thresholds defining the nodes in the decision tree.

The size of the tree, its complexity (reflecting predictors' interactions), the number of observations in the terminal nodes and the criteria to stop the process are defined as model hyper-parameters and form the basis of more complex designs.

### 3.1. Boosted Regression Trees (BRT)

Boosting Regression Trees (BRT) or Gradient Boosting Trees (GBT) are based on the building of a large number of small decision trees. The boosting aspect refers to fitting repeatedly very simple and basic classifiers, in which a different subset of the data is used for fitting at each iteration [9]. The Gradient technique is used to reduce the variance in the model; sequentially, each new tree added to the model is fitted to explain the remaining variance from the previous trees.

While BRT is considered a robust ML method, including its use for spatiotemporal analyses [19,20,22,24], it is known as having a tendency to overfit, unless a very large amount of data is available [25].

### 3.2. Random Forest (RF)

Random Forest (RF), first described by Breiman in 2001 [26], consists of a large collection of decision trees [9]. To grow an RF tree, random inputs and predictors are selected at each node [26], and this randomness is thought to reduce overfitting. RF is also considered a robust ML method that can handle outliers and noise while being faster than bagging- and boosting-based methods [26].

RF is not explicitly designed to explore spatial observations [15], and is known to produce suboptimal prediction when sampling is spatially biased and/or in the presence of strong spatial correlation [15]. However, spatiotemporal resampling strategies and variable selection processes have been developed to overcome this challenge [16,27].

## 4. Models' workflow

In order to assess the importance of integrating uncertainty on the response variable, we implemented two approaches The former relies on generating and assessing model predictions using the ***median*** estimates of the FoI for each serosurvey as an outcome (referred to as MedFoI). The latter seeks to propagate the uncertainty linked to the catalytic model-derived 'observations' by accounting for the ***full posterior distribution*** of the FoI estimates (referred to as FullPostFoI).

With the MedFoI approach, models are fitted to the median FoI estimates and the performance indicator, the predictive $R^2$, is based on central tendencies only. With the FullPostFoI approach, models are fitted on the full posterior distribution of FoI estimates and the performance indicator is based both on central tendency and on the amount of overlap between the 'observed' and predicted distribution of the outcome. This allowed us to quantify the ability of the models to match the uncertainty surrounding the FoI estimates (i.e. the outcome) inherited from the catalytic model (Fig 1). The percentage of overlap was obtained using the "overlap" function from the Overlapping R-package [28] and provided the proportion of the area of two kernel density estimations that overlap [29].

For the two approaches, we defined six different coefficients of determination ($R^2$) linked to the sampling strategy. An $R^2$ was estimated based directly on the data used to fit the models; a predictive $R^2$ was estimated based on a proportion of the dataset that was not used to fit the models, i.e. the cross-validation (CV) set. In addition, both urban- and rural-specific predictive $R^2$ were estimated based on the urban/rural data from the CV set. Finally, in the ML frameworks, a resample $R^2$ was estimated based on out-of-sample data for each resample iterations (see Fig 2 for a schematic representation of these approaches).

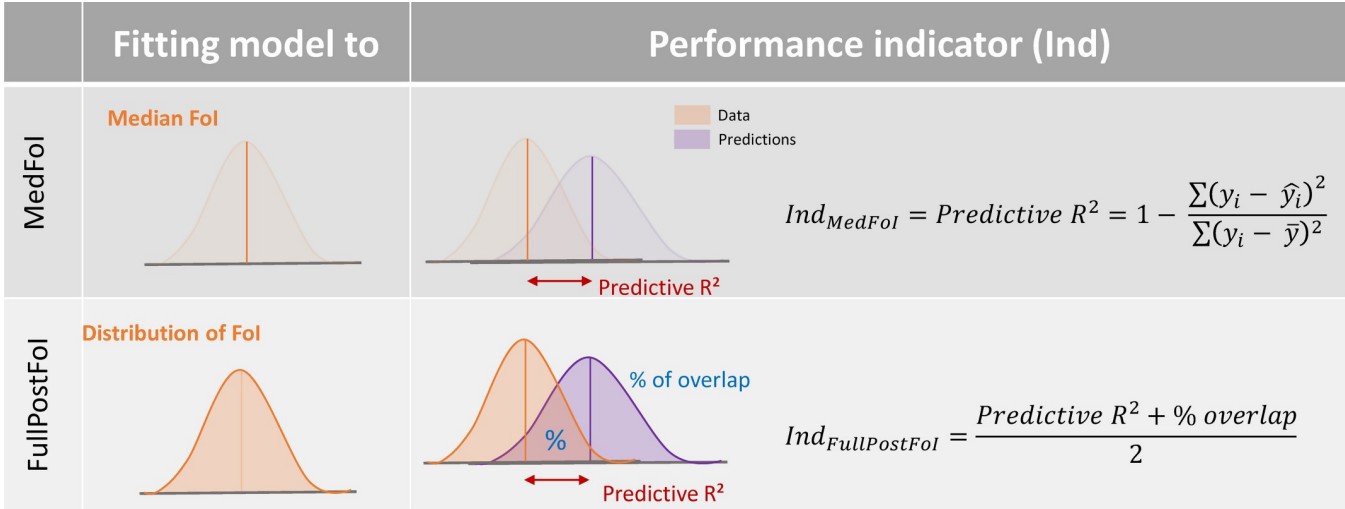

**Fig 1. Graphical representation of the two modelling approaches used for each of the frameworks tested.** The upper panel corresponds to MedFoI (fitted on median FoI); the lower panel depicts the FullPostFoI (fitted on full posterior FoI). The predictive $R^2$ values are calculated on cross-validation sets for both approaches (see Fig 2). In the upper panel, the performance indicator, *Ind*, is the $R^2$, based on central tendency alone; in the lower panel, both central tendency and percentage of overlap enter into the calculation of the performance indicator, *Ind* (as the arithmetic mean between $R^2$ and percentage overlap). The percentage of overlap (% of overlap) represents the proportion of the 'observed' and predicted distributions that overlap.

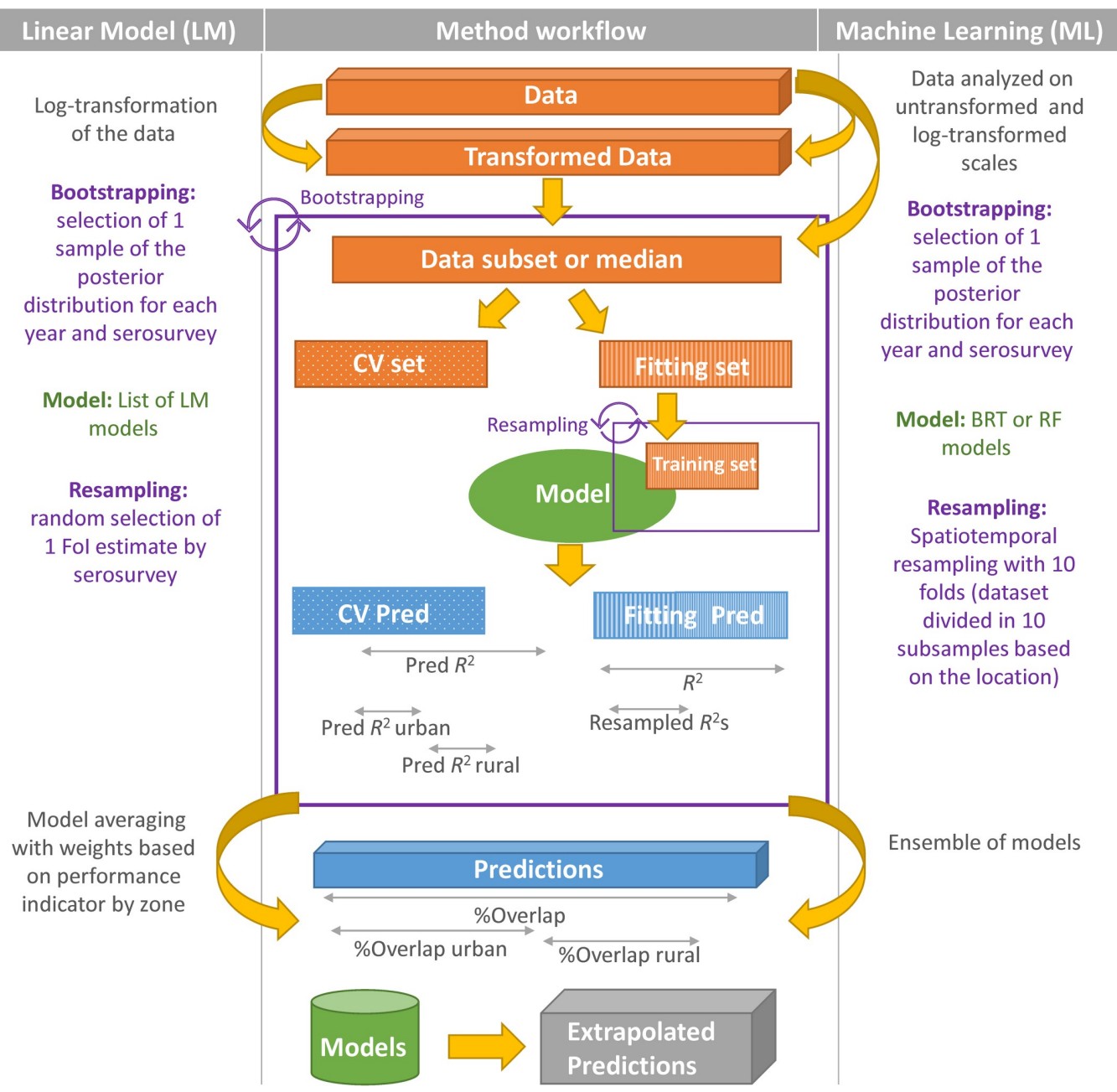

**Fig 2. Description of the modelling workflow for the Linear Models (LM) and the Machine Learning (ML) frameworks.** ML framework include Boosted Regression Trees (BRT) and Random Forest (RF) methods. CV denotes cross-validation; Pred $R^2$ urban and Pred $R^2$ rural denote urban- and rural-specific predictive $R^2$ values that were estimated based on the urban/rural data from the CV set; %Overlap indicates the proportion of the 'observed' and predicted distributions that overlap (see Fig 1), assessed over all settings and for urban and rural settings separately.

While the LM framework necessitated transformation of the data to normalise them, ML methods should, in principle, to be able to handle data without requiring normalisation (i.e. without requiring that their distribution is Gaussian). However, this process can help improve the performance of the model and was, therefore, tested (i.e. ML approaches were used to predict the FoI values both on non-transformed and log-transformed scales).

While the LM framework relies on a list of plausible and pre-defined linear models including interactions between factors (predictors) and excluding correlated predictors, the ML framework is implemented in two steps, to be fitted only on the ten most important variables. At first, ML models were fitted using all the predictors available, then the importance/influence of each predictor was assessed, and the 10 most influential factors were used in the second step, during which the models were fitted again, and predictions extracted.

Finally, ML requires a tuning step, during which the best hyperparameters are selected. Also, the resampling strategy used within the LM framework relies on random resampling while a spatiotemporal resampling strategy has been used for the ML framework. A detailed description of the tuning of hyperparameters and the comparison of several resampling strategies is available in S1 Appendix of the Supporting Information File, including details about the tuning of hyperparameters and the comparison of several resampling strategies.

## 5. Indicators used to compare LM with ML frameworks

The best ML models obtained were then compared with the LM framework previously developed [8] in terms of their performance indicators, predictions, and ability to propagate uncertainty. In addition to these aspects, the models' ability to deal with temporal and spatial correlation, as well as their different computational aspects entered the comparison.

To allow comparison of predictive ability across multiple serosurveys, the distributions of predictions were standardised to the 'observations', allowing us to visualise whether the median and confidence intervals of the predictions matched those (median and credible intervals) of the catalytic model-derived FoI 'observations'. This process was performed at the serosurvey level to assess how much of the uncertainty inherited from the FoI calculation (via catalytic model fitting) was propagated into the predictions (see S2 Appendix in the Supporting Information File for details).

The uncertainty in the predictions was quantified using the Coefficient of Variation based on the standardised Median Absolute Deviation (MAD-CV), as FoI values were not normally distributed [30]. (Note that MAD-CV refers to coefficient of variation, whilst CV denotes cross validation.)

The residual spatial correlation was assessed using the Moran's $I$ heterogeneity test from the "spdep" R package [31]. For the LM framework, the Moran's $I$ test was applied on all the residuals (originating from the cross validation (CV) and fitting sets) excluding those from the rural–urban mixed settings (as LM model selection was based on setting type and no model was explored, selected or averaged for mixed settings, and thus no predictions were produced for the 'observed' FoI values corresponding to such settings). For ML models, the Moran's $I$ test was applied to the residuals of the CV set. Residuals for a single year were used to exclude potential temporal autocorrelation, and for presentation purposes, we selected 2005 as the year with the largest number of independent FoI 'observations'.

The residual temporal correlation was tested using a Durbin-Watson test (DW) [32] (see Eq 1 for the DW statistic). In order to capture the residual correlation inherited from the estimation of the FoI values through fitting the catalytic model, the residuals being compared were always from the same serosurvey and for consecutive years. Thus, the DW statistic provided the residual serial correlation for a lag of one year,

$$DW = \frac{\sum_{i=0}^{n} \left(r_i - lag(r_i)\right)^2}{\sum_{i=0}^{n} r_i^2} \qquad \text{Eq1}$$

where $r$ denotes the residuals for $i$ serosurveys, $lag$ is one year (for consecutive, yearly series of serosurveys), and $n$ the number of 'observations' tested.

**Table 1. Median cross-validation performance values for the two Machine Learning modelling methods investigated.**

| | BRT | | | | | | RF | | | | | |
| --- | --- | --- | --- | --- | --- | --- | --- | --- | --- | --- | --- | --- |
| | All settings | | Urban | | Rural | | All settings | | Urban | | Rural | |
| | non | log | non | log | non | log | non | log | non | log | non | log |
| *MedFoI* | | | | | | | | | | | | |
| $R^2$ (%) | 98 | 95 | 95 | 96 | 94 | 90 | 98 | 96 | 90 | 98 | 93 | 93 |
| Overlap | 23 | 19 | 21 | 19 | 25 | 19 | 22 | 21 | 20 | 21 | 25 | 21 |
| *FullPostFoI* | | | | | | | | | | | | |
| $R^2$ (%) | 60 | 58 | 53 | 68 | 39 | 68 | 63 | 59 | 63 | 68 | 69 | 70 |
| Overlap | 25 | 36 | 24 | 34 | 22 | 36 | 40 | 42 | 42 | 42 | 40 | 42 |
| Indicator | 43 | 48 | 37 | 52 | 25 | 52 | 51 | 50 | 50 | 55 | 52 | 55 |

*MedFoI*: models fitted on median FoI; *FullPostFoI*: models fitted on full posterior FoI, without (non) or with log-transformation (log) of Force-of-infection (FoI) 'observations' (generated by fitting catalytic models to age-stratified serological surveys for Chagas disease in Colombia, with yearly-varying FoI [8]).

BRT: Boosted Regression Trees; RF: Random Forest methods; performance indicators are reported for either all settings (urban, rural, indigenous and mixed), urban, or rural settings separately. The predictive $R^2$ values were calculated on cross-validation datasets and are expressed as percentage.

Overlap: proportion (expressed as percentage) of 'observed' and predicted distributions that overlaps (reflective of the degree of dispersion around central tendency).

For *MedFoI* models, the performance indicator is the value of $R^2$ alone. Therefore % overlap is presented for comparison but was not used in fitting or selecting models.

For *FullPostFoI*, the performance indicator is the arithmetic mean between $R^2$ and % of overlap (see Fig 1).

All ML analyses were run under the mlr3 framework (an object-oriented machine learning framework in R) [33] using R-4.0.3 software [34].

## Results

### 1. Comparison of the performance of LM and ML frameworks

The predictive $R^2$ values for the LM framework obtained, on average, for its 5 best-fitting models, were 77% and 70%, with %overlap of 54% and 39% for urban and rural settings, respectively [8]. For the ML frameworks, the MedFoI approach yielded substantially better predictive $R^2$ values (ranging between 90% and 98%), but the degree of overlap between the distributions of the FoI 'observations' and the predictions was substantially lower (19%–25%), reflective of a tighter distribution around the central estimates and thus indicating over-confidence in the predictions when using such a simple approach (i.e. an approach that ignores the uncertainty linked to the outcome). The FullPostFoI approach gave a more balanced performance indicator, with predictive $R^2$ values ranging between 39% and 70%, and %overlap between 22% and 42% (Table 1). For both BRT and RF methods, the use of log-transformation to normalise the distribution of the FoI 'observations' consistently led to improved results (Table 1), with predictive $R^2$ values ranging between 59% and 70%, and %overlap between 34% and 42%.

Nested resampling, tested on the RF method with log-transformation, did not substantially improve model performance. Thus, the following subsections focus on the results obtained by fitting the frameworks on the full posterior distribution of the log-transformed FoI.

### 2. Comparison of the influence of predictor variables

The factors selected for the ML models were consistent with those that had been selected for the LM framework (Table 2); a Spearman correlation test showed that there was substantial rank correlation of the predictors included among the three models investigated (with Spearman rank correlation coefficient, $r_S$, between LM and BRT = 0.50; between LM and RF = 0.54, and between BRT and RF = 0.64, all p-values <0.05).

**Table 2. Standardized relative influence, importance and rank of the predictors included in Boosted Regression Trees (BRT) and Random Forest (RF) Machine Learning models and normalised number of times the predictors were used in the linear model (LM) framework and their rank when using the full posterior distribution of FoI estimates.**

| Predictors | | BRT | | RF | | LM | |
|---|---|---|---|---|---|---|---|
| Code | Name | Influence | Rank | Importance | Rank | Used | Rank |
| | **Serosurvey characteristics:** | | | | | | |
| S01 | Year of the survey | 0.20 | 2 | 0.17 | 1 | 0.05 | 4 |
| S02 | Rural setting | 0.03 | 11 | 0.03 | 11 | 0.14 | 2 |
| S03 | Urban setting | 0.03 | 10 | 0.04 | 7 | 0.15 | 1 |
| S04 | Indigenous setting | 0.20 | 1 | 0.12 | 4 | 0.15 | 1 |
| S05 | Latitude | 0.14 | 3 | 0.16 | 2 | 0.14 | 2 |
| S06 | Longitude | 0.04 | 8 | 0.09 | 5 | 0.02 | 7 |
| | **Blood-bank data:** | | | | | | |
| B01 | Seroprevalence | 0.00 | NU | 0.03 | 14 | 0.04 | 5 |
| B02 | Proportion of blood units screened | 0.00 | NU | 0.03 | 10 | 0.01 | 8 |
| | **Demography:** | | | | | | |
| D01 | Population density | 0.10 | 4 | 0.13 | 3 | 0.01 | 8 |
| D02 | Poverty | 0.07 | 6 | 0.01 | 15 | 0.04 | 5 |
| D03 | Rural Indigenous Population size | 0.00 | NU | 0.03 | 9 | 0.00 | NU |
| | **Climate:** | | | | | | |
| | *Continuous* | | | | | | |
| C01 | Polar climate frequency | 0.03 | 12 | 0.00 | 18 | 0.04 | 5 |
| C02 | Tropical climate frequency | 0.03 | 9 | 0.04 | 8 | 0.01 | 8 |
| C03 | Temperate climate frequency | 0.04 | 7 | 0.00 | 17 | 0.00 | NU |
| C04 | Arid climate frequency | 0.00 | NU | 0.00 | 21 | 0.00 | NU |
| | *Categorical* | | | | | | |
| C05 | Tropical climate categorised | NT | | | | 0.06 | 3 |
| C06 | Polar Climate Presence | 0.00 | NU | 0.00 | NU | 0.00 | NU |
| | **Entomological data:** | | | | | | |
| | *At departmental level* | | | | | | |
| V01 | *R. prolixus* geographical extent | 0.00 | NU | 0.06 | 6 | 0.02 | 7 |
| V02 | *T. dimidiata* geographical extent | 0.00 | NU | 0.03 | 12 | 0.01 | 8 |
| V03 | *R. prolixus* presence | 0.00 | NU | 0.00 | NU | 0.00 | NU |
| V04 | *T. dimidiata* presence | 0.00 | NU | 0.00 | NU | 0.03 | 6 |
| | *At municipality level* | | | | | | |
| V05 | *R. prolixus* density | 0.00 | 13 | 0.00 | NU | 0.01 | 8 |
| V06 | *T. dimidiata* density | 0.00 | 15 | 0.00 | 16 | 0.01 | 8 |
| V07 | *R. prolixus* presence | 0.00 | NU | 0.00 | NU | 0.00 | NU |
| V08 | *T. dimidiata* presence | 0.00 | NU | 0.00 | 20 | 0.00 | NU |
| | **Interventions:** | | | | | | |
| | *At municipality level* | | | | | | |
| I01 | Intervention intensity | 0.00 | 14 | 0.00 | NU | 0.00 | NU |
| I02 | Intervention category | NT | | NT | | 0.01 | 8 |
| | *At household level* | | | | | | |
| I03 | Household intervention | 0.00 | NU | 0.00 | NU | 0.00 | NU |
| I04 | Household intervention category | NT | | NT | | 0.01 | 8 |
| | **Temporal factors:** | | | | | | |
| T01 | Year | 0.09 | 5 | 0.03 | 13 | 0.01 | 8 |

*(Continued)*

**Table 2.** (Continued)

| Predictors | | BRT | | RF | | LM | |
|---|---|---|---|---|---|---|---|
| Code | Name | Influence | Rank | Importance | Rank | Used | Rank |
| T02 | Decade | 0.00 | NU | 0.00 | NU | 0.00 | NU |

NU: Not used in the model; NT: not tested in the model.

R. prolixus: Rhodnius prolixus; T. dimidiata: Triatoma dimidiata.

The shade of green is associated to the rank of the predictors with darker green predictors having more importance.

The type of setting was the most important factor for both the LM and BRT. Latitude, year when the serosurvey was conducted, and population density also played an important role. Poverty, climatic and entomological features had a moderate role.

For the ML frameworks, blood-bank and intervention-related features were less influential than for the LM framework. Generally, the year (temporal trend) seemed to play a greater role in the ML models.

### 3. Comparison of spatial trends in predictions

All methods showed generally similar spatial trends and comparable levels of uncertainty (Fig 3) for FoI prediction across Colombia (using the year 1990 for illustration as the pattern is consistent in time). Generally, FoI estimates were higher in northern and eastern municipalities and lower in the south of the country, with the latter being associated with higher uncertainty (Fig 3). The BRT framework predictions showed increased spatial heterogeneity, while predictions from the LM framework resulted in more spatially uniform predictions.

When comparing FoI predictions directly across the three methods, for urban and rural settings (Fig 4), we found good agreement between all of them, particularly between RF and LM. Generally, the BRT tended to predict higher FoI values in both settings. The patterns observed in the entire country seemed to follow what was observed in the catchment areas (municipalities where at least one serosurvey was conducted).

### 4. Comparison of temporal trends in predictions across serosurveys

When comparing 'observations' with predictions over time, all methods performed well regarding their ability to capture central trends (Fig 5). However, the LM framework was better at capturing uncertainty, as the confidence bounds of the predictions mirrored more closely the credible intervals (CrI) of the 'observations'.

The median uncertainty across municipalities (Table 3) was comparable using any of the methods and restricting the assessment to 'in catchment area' only (i.e. municipalities where at least one serosurvey had been conducted) or not ('out of catchment area'). However, for some municipalities, the uncertainty associated with the LM framework increased dramatically.

Comparatively, the RF method produced more uniform uncertainty across predictions, with median and range similar to those yielded by the other (BRT and LM) methods, but with fewer municipalities with substantial uncertainty (defined as MAD-CV>2), and only a moderate number of municipalities with extreme uncertainty (defined as MAD-CV>5).

### 5. Residual spatial and temporal correlation

While the ML-based methods did not show any significant spatial correlation in their residuals, this was not the case with the LM framework (Table 4). For all models, the DW test's statistic (see Eq 1) showed a significant residual temporal correlation between residuals from the

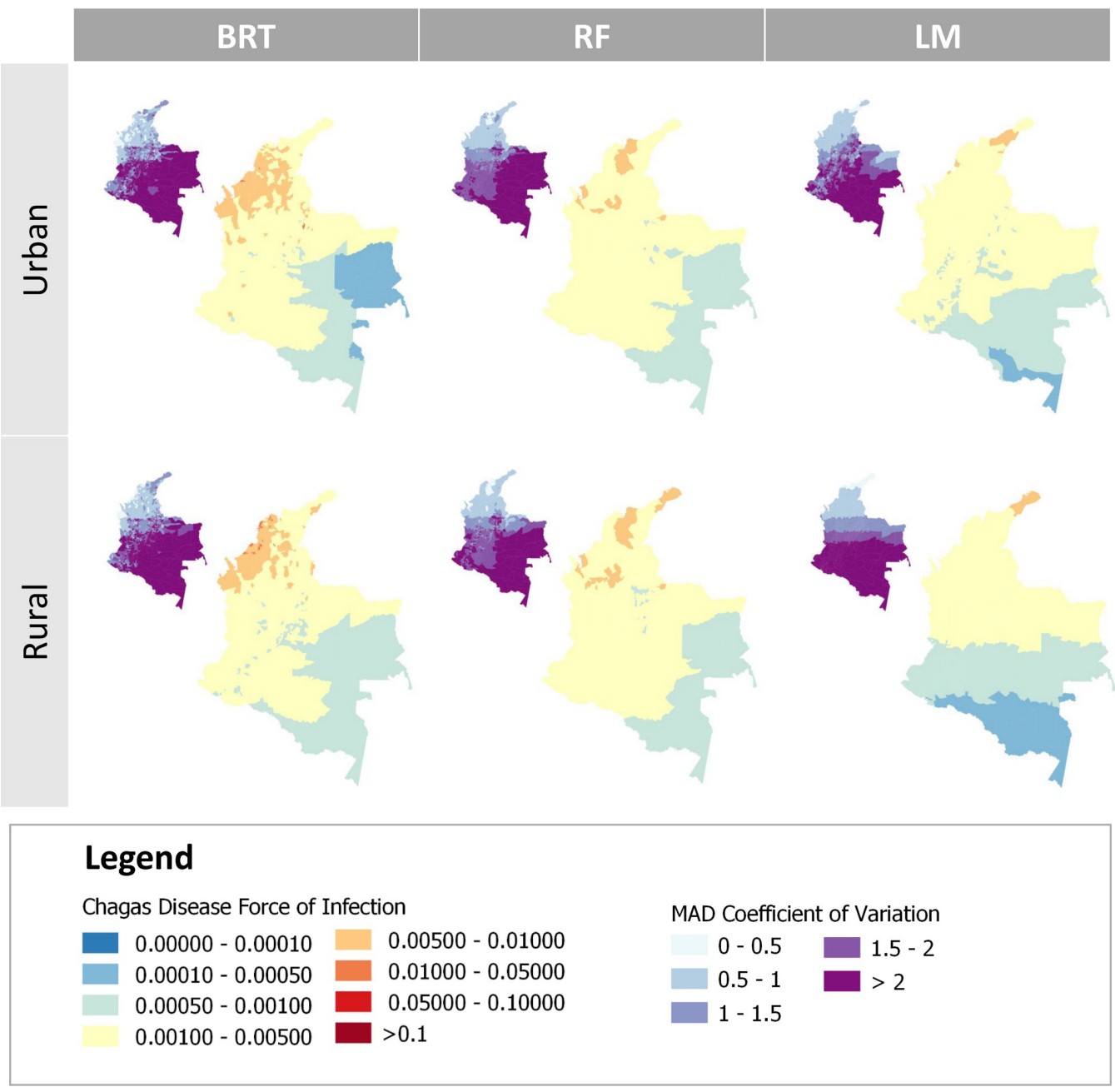

**Fig 3. Spatial distribution of the Force-of-Infection of Chagas Disease (per year and per susceptible individual), in Colombia.** The predicted distribution was generated using two Machine Learning (Boosted Regression Trees (BRT) and Random Forest (RF)) methods and a Linear Model (LM) framework (main maps); the associated uncertainty (small map insets) presents the Median Absolute Deviation (MAD) Coefficient of Variation (MAD-CV). Predictions were obtained at the municipality level for urban and rural settings, in 1990. (Borders shapefile at the ADM2 level obtained from GADM, https://gadm.org/download_country.html).

same serosurvey, with a stronger effect for serosurveys conducted in indigenous settings (S2 Fig in Supporting Information File).

## 6. Computational aspects

Computationally, RF and BRT required the least effort (31 and 42 hrs respectively, on standard laptop, with an i7-8565U processor and 16.0 GB RAM) (Table 5). By contrast, although

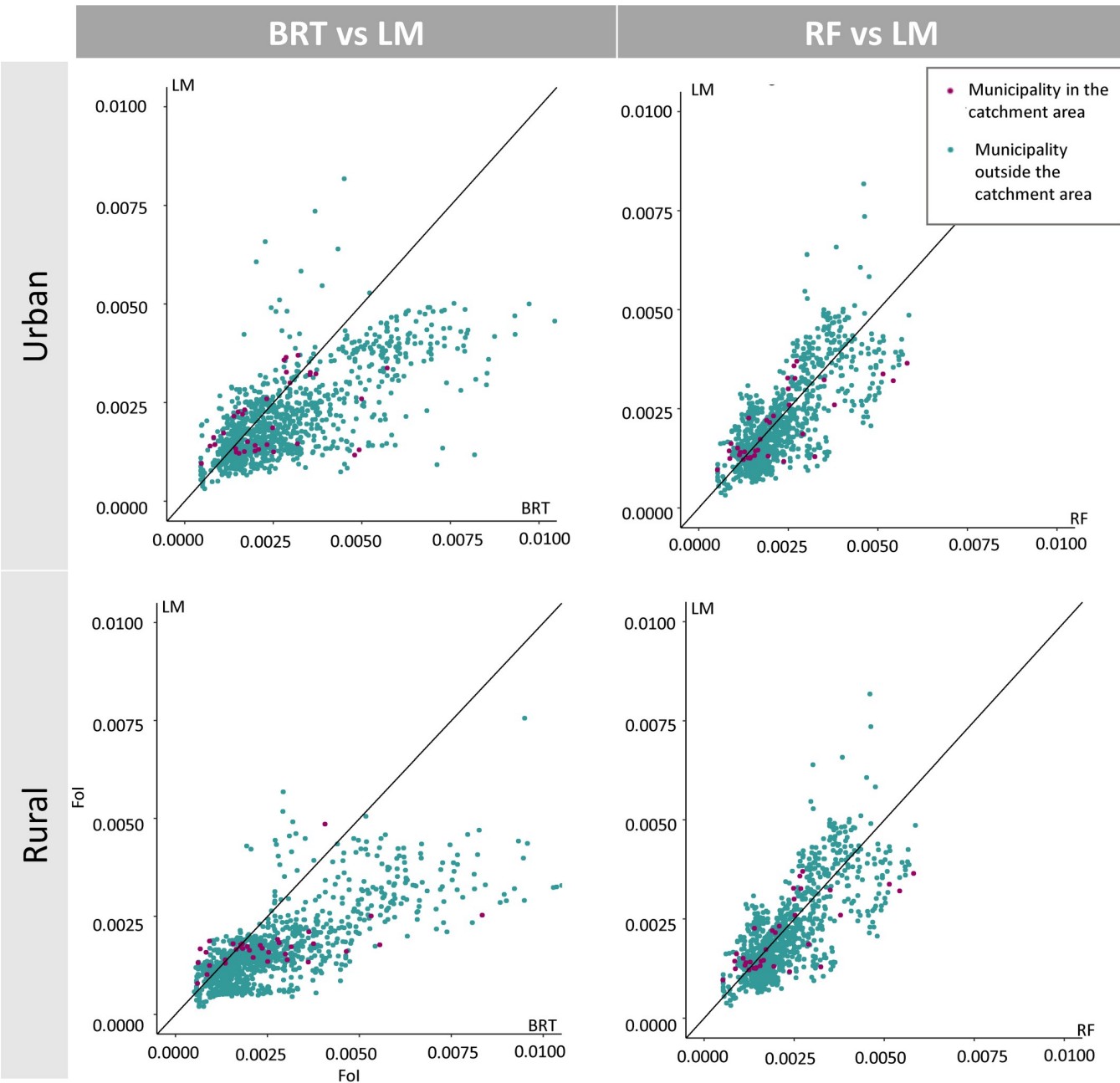

**Fig 4. Comparison of predicted Chagas disease Force-of-Infection (FoI) values for urban or rural settings at municipality level, in Colombia for the year 1990.** The values were obtained by two Machine Learning (Boosted Regression Trees (BRT) and Random Forest (RF)) methods and a Linear Model (LM) framework using log-transformed FoI estimates from the FullPostFoI approach (see Models' workflow subsection in Methods and Fig 1 for a description of this approach). The upper panel presents the results for urban settings; the lower panel presents the results for rural settings. Purple-coloured dots denote municipalities where at least one serosurvey had been conducted ('in catchment area'); teal-coloured dots denote municipalities where no serosurveys had been conducted or were not included in our analyses ('outside catchment area'). The black solid diagonal line represents perfect agreement between the two frameworks being compared.

implementation of LM required far fewer R-packages than the ML framework, it took over twice the time to run when compared to RF (72 hr). Also, the computer hard-drive space that was required to store 'objects' and model outputs was about 20 times higher for LM than for

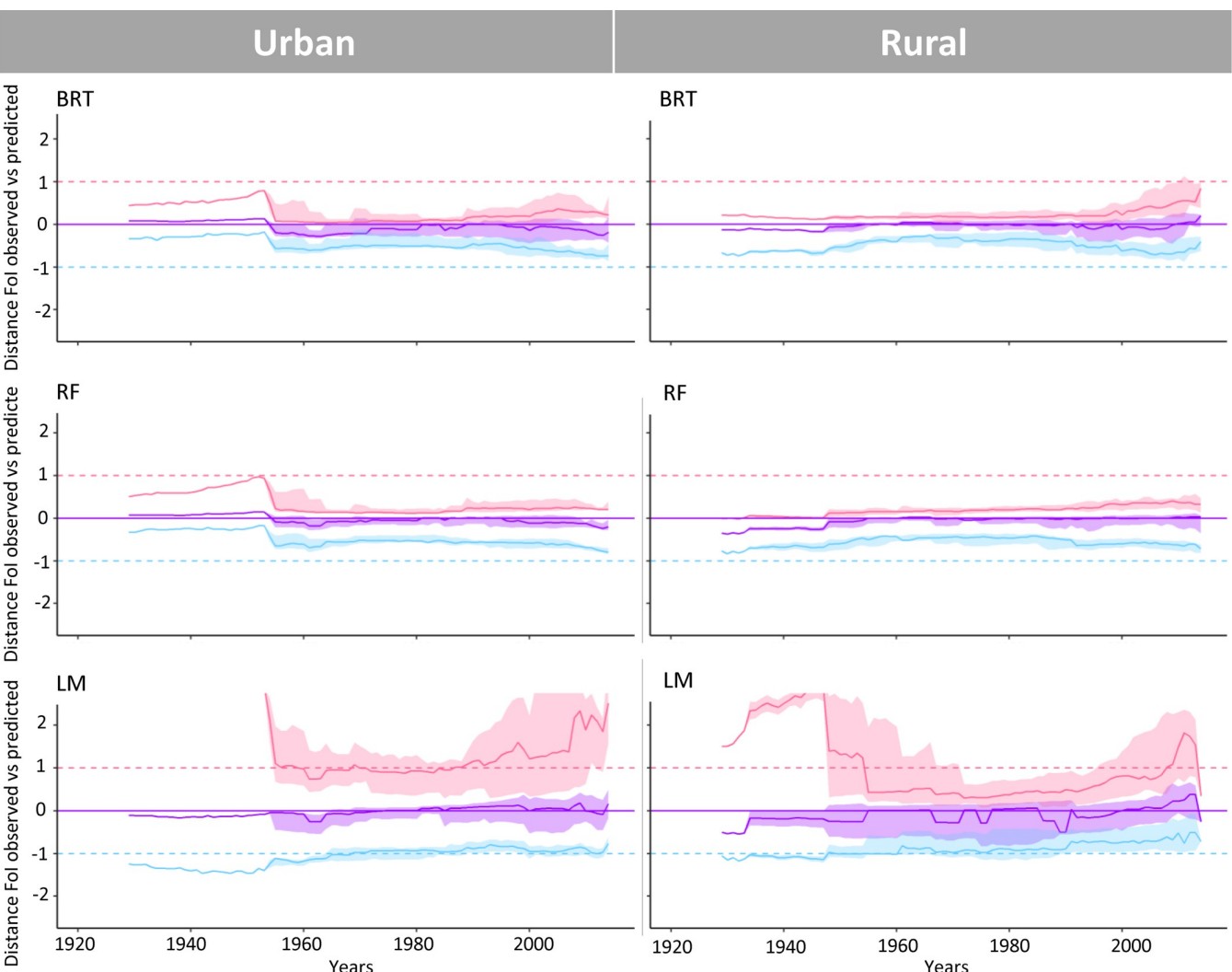

**Fig 5. Standardised comparisons of 'observed' and predicted distributions across serosurveys and by setting type.** Comparisons were made for two Machine Learning (Boosted Regression Trees (BRT) and Random Forest (RF)) methods (upper and middle panels) and a Linear Model (LM) framework (lower panel) using log-transformed (log) Force-of-Infection (FoI) estimates from the FullPostFoI approach for urban and rural Chagas disease settings in Colombia across 9 decades. The solid lines and envelopes show standardised distances between FoI 'observations' and predictions, with purple-colour lines representing the median, and the pink and blue lines representing, respectively, the upper and lower bounds of the 95%CrI. If medians and 95% confidence bounds of the predictions matched exactly the corresponding measures for all the 'observations' across serosurveys, then the solid and dashed lines would fully overlap.

the ML framework. Finally, the overall implementation of the models was substantially simpler for the ML framework; particularly to make adjustments and updates.

## Discussion

Our comparative analyses indicated generally consistent results among the three modelling methods investigated to generate Chagas disease FoI predictions, namely, the linear model (LM) framework we previously developed [8], and the two Machine Learning (ML) methods explored here (Boosted Regression Trees (BRT) and Random Forest (RF)). The predictors that were selected, as well as the location of the most uncertain FoI values were coherent and generally consistent among the three methods (Table 2 and Figs 3 and 4). Not entirely surprising, RF was faster to run than BRT and LM [26] (Table 5).

**Table 3. Uncertainty across Chagas disease Force-of-Infection predictions for the three frameworks under comparison.** The uncertainty was estimated using the Median Absolute Deviation Coefficient of Variation (MAD-CV) of the predictions for Colombia in 1990, in (urban and rural) areas where at least one serosurvey had been conducted ('in catchment area') and where no data were available or used in the analyses ('out of catchment area'). The number of municipalities where MAD-CV is greater than 2 (substantial uncertainty) or greater than 5 (extreme uncertainty) is also included.

| | MAD-CV values (range) | | | | Number of municipalities MAD-CV> 2 | | Number of municipalities MAD-CV> 5 | |
| | In catchment area | | Out of catchment area | | | | | |
| | Urban | Rural | Urban | Rural | Urban | Rural | Urban | Rural |
|---|---|---|---|---|---|---|---|---|
| BRT | 1.45 (0.31–7.16) | 1.54 (0.39–5.40) | 1.48 (0.31–7.41) | 1.48 (0.17–6.32) | 338 | 335 | 25 | 24 |
| RF | 1.47 (0.48–5.28) | 1.45 (0.40–5.29) | 1.48 (0.47–5.24) | 1.49 (0.44–5.22) | 145 | 198 | 10 | 8 |
| LM | 1.60 (0.70–2.73) | 1.29 (0.44–2.76) | 1.48 (0.32–8.19) | 1.50 (0.24–11.00) | 284 | 266 | 6 | 11 |

BRT: Boosted Regression Trees; RF: Random Forest; LM: Linear Model.

Based on the performance indicators used, RF performed best (Table 1) but did less well when considering the propagation of uncertainty in the FoI inherited from the catalytic model (Fig 5). Also, RF generated fewer municipality-level predicted values with substantial or extreme uncertainty (Table 3). All methods, when fitted on the median FoI alone (MedFoI approach), were unable to capture the uncertainty in the response variable (the FoI 'observations' generated by fitting the catalytic model to the age-stratified serosurveys), leading to over-confident predictions (with high predictive $R^2$ values but smaller % of overlap values). This highlights an important issue not fully addressed in the literature, as most publications using FoI data to infer spatiotemporal patterns of infectious disease incidence tend to use the central FoI estimates alone to fit predictive models (i.e. using what we labelled here as the MedFoI approach). We argue that neglecting to appreciate and propagate the uncertainty inherent in their estimation [5–7] may therefore lead to significant over-confidence in predictions. This issue, already highlighted in our previous LM work [8], is not mitigated by implementing ML frameworks, and deserves careful consideration, not only from a methodological perspective, but importantly, when the results are applied in policy-relevant contexts [35].

Indeed, quantifying and communicating uncertainty in FoI appropriately is critical when the results of predictive models are used to inform stakeholders and public health programme managers on the level of certitude associated with exposure risk or number of cases. Thus areas/populations for which exposure has been certainly high or low can be differentiated from those with exposure levels or number of cases that necessitate further investigation due to highly uncertain estimation.

Even when the three methods showed good performance and generally good agreement at the serosurvey level (Fig 4), the residuals remained correlated in time (Table 4). Thus, the

**Table 4. Spatial and temporal correlation test statistics and statistical significance of the spatial correlation test applied to the cross-validation residuals for the two Machine Learning (BRT, RF) and the Linear Model (LM) methods under consideration.**

| | BRT | RF | LM† |
|---|---|---|---|
| **Spatial correlation test:** | | | |
| Moran's $I$ statistic | 0.00 | 0.00 | 0.06* |
| **Temporal correlation test:** | | | |
| DW statistic | 0.06* | 0.04* | 0.00* |

BRT: Boosted Regression Trees; RF: Random Forest; LM: Linear Model.

DW: Durbin-Watson statistic (see Eq 1).

†See methods for calculation of the LM residual correlation.

*p-values significant et 5%.

**Table 5. Comparison of computational aspects for the Machine Learning (Boosted Regression Trees (BRT), Random Forest (RF)) and Linear Model (LM) methods investigated.** The methods under comparison used log-transformed FoI values from the FullPostFoI approach.

|  | BRT | RF | LM |
|---|---|---|---|
| Number of R packages needed | 20 | 20 | 6 |
| Time required for models to run (hr) | 42.5 | 31.0 | 72.0 |
| Hard-drive space requirements (MB) | 149 | 114 | 2,048 |

BRT: Boosted Regression Trees; RF: Random Forest; LM: Linear Model.

hr: hours; MB: Megabytes.

correlation inherited from the FoI calculation was not fully accounted for in any of our methods, i.e. none of the predictors included was able to account for the full extent of this correlation.

While the final ML models showed no evidence of residual spatial correlation (Table 4), the spatial extrapolation shown (Fig 3) should be interpreted with caution, as the (ground-truth) serosurveys available had only been conducted in a relatively small number of municipalities and tended to be aggregated in the same area (S1 Fig). When using RF, the degree of uncertainty inside and outside 'catchment areas' was consistent, suggesting reliable extrapolation. This contrasted with the LM framework, which predicted large uncertainty in some municipalities.

Most of the earliest serosurveys (up to early 2000) seemed to have targeted high-risk populations [36], presumably because the perceived risk of Chagas disease transmission in those areas was higher and required improved situational awareness. However, using only such information to make predictions across Colombia would have led to higher predicted FoI in areas where no ground-truth data had been collected. By contrast, the most recent serosurveys (2010–2014) seem to have been conducted on more representative samples of the population, presumably motivated by providing a more realistic assessment of the epidemiological situation nationally and demonstrating progress in reducing vectorial transmission. We, therefore, included the year when the serosurvey took place to account for this bias and, in fact, this variable appeared to be one of the most influential in all three methods and particularly for BRT and RF (Table 2). This demonstrates the crucial importance of understanding the motivation behind the implementation of serosurveys in order to assess the sampling strategy and ultimately quantify potential biases that may interfere with the representativeness of FoI estimates.

Finally, and regarding computational aspects, the LM framework required substantial user-input to prepare the data for model fitting (including data transformation; choice of predictors included in each model; tests for spatial and temporal correlation, etc.). In contrast, ML frameworks were faster (particularly RF) and required less pre-processing of the data and hard-drive space (Table 5). These features render the ML models more flexible, more readily updatable, and thus easier and simpler to be extended to other Chagas disease-endemic countries, and potentially to other infectious diseases, including neglected tropical diseases, for which serological surveys are regularly conducted as surveillance tools to assess epidemiological situation, incidence, and impact of control interventions across spatial and temporal scales [5,6,37,38].

## Concluding remarks

ML methods are increasingly used to derive computationally efficient algorithms for data analysis that are agnostic to the distributional properties of such data. They represent an attractive

modelling tool for the generation of predictive maps of important infectious disease epidemiological metrics, such as the FoI. Most published literature on the subject use measures of FoI central tendency, neglecting to quantify, propagate and ultimately communicate the uncertainty appropriately. We show that the uncertainty on the input variables cannot be neglected whatever statistical method is used and furthermore that the choice of modelling framework requires careful consideration according to the ultimate objectives of the modelling endeavour. If the aim is, for instance, to use the predicted FoI patterns to provide numbers of cases and estimates of the associated disease burden, ML framework (and particularly RF) would indeed be an optimal choice, as capturing the median (central tendency behaviour) may be sufficient and computationally efficient. However, if the objective is to identify areas where serological surveillance surveys are scarce and should be conducted to improve the reliability of FoI estimates and provide ground-truth data, we conclude that the LM framework, albeit more time-consuming and computationally intensive, would provide a better indication of where uncertainty is greatest. Although in this paper we focused on Chagas disease in Colombia as a case study, the modelling frameworks compared here can be applied to other Chagas disease-endemic countries and to infectious diseases (including neglected tropical diseases) for which age-stratified serological data are regularly collected.

## Supporting information

**S1 Appendix. Machine Learning tuning.**
(DOCX)

**S2 Appendix. Comparing observations and predictions across serosurveys.**
(DOCX)

**S1 Table. Variables tested as factors in the geospatial analyses of Chagas disease in Colombia.**
(DOCX)

**S1 Fig. Locations and sample sizes of Chagas disease serosurveys conducted in Colombia with information on the location at the municipality level, from 1998 to 2014 (Borders shapefile at the ADM2 level obtained from GADM, https://gadm.org/download_country. html).** Figure extracted from (2). The grey boundaries delimitate the Departments.
(TIF)

**S2 Fig. Comparison of residual serial correlation for Boosted Regression Trees (BRT), Random Forest (RF) and Linear Model (LM) models built on the log scale based on the FullPostFoI approach.** Each line corresponds to one serosurvey.
(TIF)

## Acknowledgments

The authors thank the Pan American Health Organization (PAHO) and in particular Dr Luis Gerardo Castellanos for their support. Our thanks also go to Dr Alpha Forna for valuable discussions.

## Author Contributions

**Conceptualization:** Julia Ledien, Zulma M. Cucunubá, Gabriel Parra-Henao, Eliana Rodríguez-Monguí, Andrew P. Dobson, Susana B. Adamo, María-Gloria Basáñez, Pierre Nouvellet.

**Data curation:** Julia Ledien, Zulma M. Cucunubá, Gabriel Parra-Henao, Eliana Rodríguez-Monguí, Pierre Nouvellet.

**Formal analysis:** Julia Ledien.

**Funding acquisition:** Zulma M. Cucunubá, Pierre Nouvellet.

**Investigation:** Julia Ledien, Zulma M. Cucunubá, Gabriel Parra-Henao, Eliana Rodríguez-Monguí, Andrew P. Dobson, Susana B. Adamo, María-Gloria Basáñez, Pierre Nouvellet.

**Methodology:** Julia Ledien, Zulma M. Cucunubá, Gabriel Parra-Henao, Eliana Rodríguez-Monguí, Andrew P. Dobson, Susana B. Adamo, María-Gloria Basáñez, Pierre Nouvellet.

**Supervision:** Zulma M. Cucunubá, María-Gloria Basáñez, Pierre Nouvellet.

**Validation:** Julia Ledien.

**Visualization:** Julia Ledien.

**Writing – original draft:** Julia Ledien.

**Writing – review & editing:** Julia Ledien, Andrew P. Dobson, María-Gloria Basáñez, Pierre Nouvellet.

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
