## [Decision Letter · Decision Letter 0]

26 Apr 2022

Dear Mrs Ledien,

Thank you very much for submitting your manuscript "Linear and Machine Learning Modelling for Spatiotemporal Disease Predictions: Force-of-Infection of Chagas Disease" for consideration at PLOS Neglected Tropical Diseases. As with all papers reviewed by the journal, your manuscript was reviewed by members of the editorial board and by several independent reviewers. The reviewers appreciated the attention to an important topic. Based on the reviews, we are likely to accept this manuscript for publication, providing that you modify the manuscript according to the review recommendations. 

Before your manuscript can be accepted. Please kindly address the concerns of Reviewer Number 2. You may accept or rebutt his/her concerns. Thank you.

Sincerely,

Uwem Friday Ekpo, PhD

Associate Editor

Robert Reiner

Deputy Editor

Before your manuscript can be accepted. Please kindly address the concerns of Reviewer Number 2. You may accept or rebutt his/her concerns. Thank you.

Reviewer's Responses to Questions

**Key Review Criteria Required for Acceptance?**

**Methods**

-Are the objectives of the study clearly articulated with a clear testable hypothesis stated?

-Is the study design appropriate to address the stated objectives?

-Is the population clearly described and appropriate for the hypothesis being tested?

-Is the sample size sufficient to ensure adequate power to address the hypothesis being tested?

-Were correct statistical analysis used to support conclusions?

-Are there concerns about ethical or regulatory requirements being met?

Reviewer #1: (No Response)

Reviewer #2: The Github repository may be private? It's not accessible using the link provided. The only repository associated with the user does not contain analysis scripts

How were correlated variables dealt with? Particularly with LM?

How were training and testing datasets chosen? Were these random or stratified across space and time?

Reviewer #3: The aim is to compare the performance of two Machine Model frameworks (Bosted Regression Tree - BRT; Random Forest - RF) with a Linear Model regression.

From the theoretical field, it is well designed and explained. The analysis is well established and methodologically corresponds to the objectives.

It is observed that the study population results from historical data and grouped at the community level, for which there is a bias in estimating that the risk of infection is distributed randomly when the result of the new infection results from environmental conditions. (housing, environmental interventions and insecticide use), of the person in terms of exposure (influenced by age, socioeconomic status and activity), vector (density, location, etc.) and parasites (density and dispersion)

It is expected that this situation will produce bias in the final results.

**Results**

-Does the analysis presented match the analysis plan?

-Are the results clearly and completely presented?

-Are the figures (Tables, Images) of sufficient quality for clarity?

Reviewer #1: (No Response)

Reviewer #2: Can the discussion of the results discuss 

1) Model performance for median FOI predictions. Also including a discussion of different predictors and how the direction and magnitude of these predictors varied between models.

2) Then discuss the overlap with posterior (ability to capture uncertainity)

Now it jumps between evaluation metrics, response variables and models and is difficult to follow. If this section is more linked to specific research questions, I think it would be easier to follow

Reviewer #3: The presentation of the results is adequate and they are shown clearly.

**Conclusions**

-Are the conclusions supported by the data presented?

-Are the limitations of analysis clearly described?

-Do the authors discuss how these data can be helpful to advance our understanding of the topic under study?

-Is public health relevance addressed?

Reviewer #1: (No Response)

Reviewer #2: What are the key outcomes of this research?

How applicable is this to other settings? Presumably without FOI estimates from age seroprevalence data, this would be very difficult to do.

Reviewer #3: The conclusions adequately inform the data presented, informing the limitations of the analysis as well as the diverse conditions that are presented.

The authors present their data and propose the continuation of analyzes of this type.

**Editorial and Data Presentation Modifications?**

Reviewer #1: (No Response)

Reviewer #2: (No Response)

Reviewer #3: The authors adequately express the objective of comparing the performance of two Machine Model frameworks, the results are adequately presented and in the discussion they address the different topics, clearly establishing the limitations and application of what is proposed.

**Summary and General Comments**

Reviewer #1: This work compared the ability of machine learning algorithms (here: random forest and boosted regression trees) and linear models to quantify the uncertainty around central estimates such as means or medians. The authors further compared the performance of all three modelling frameworks and their ability to predict the force of infection of Chagas disease across space and time. They have used robust age-stratified serological data from Colombia as well as relevant remotely-sensed covariates to perform all analyses. The work concluded that machine learning algorithms out-performed linear models in capturing accurate estimates for central tendency. However, linear models provided more robust estimates of uncertainty. 

This work is particularly important given the recent popularity and use of machine learning algorithms in medicine research and disease prediction. The Objectives of the study were clearly articulated and are relevant in the current sphere of disease modelling methodologies. The Methods and Results were appropriate without any ambiguities. Similar sets of data were used in all three modelling frameworks, allowing for a more complete comparison. Results were complete with ample information to aid unbiased interpretation. Authors provided plentiful important details, and often overlooked information, such as those provided in Table 5. I found this interesting. Discussion/Conclusions was balanced. This is an all-round excellent paper. I recommend that it can be published as it is, without any revision.

Reviewer #2: This paper compares the ability of ML (RF & BRT) and LM to identify key predictors of T. cruzi FOI over space and time in Colombia. Observed FOI estimates come from previous work conducted by some of the research team and used age-seroprevalence data to estimate FOI by year and municipality. Both the median FOI estimate and the full posterior distribution were used as response variables in LM and ML models. 

While it is clear a lot of work went into this paper, it’s very difficult to follow in its current form. More consistent terms throughout should help the reader follow which methods and which tests are used. For example, “FoI estimates” is used to describe the previous age-seroprevalence estimates but also the LM and ML predictions. Using these interchangeably creates unnecessary confusion and makes the key findings difficult to disentangle from previous work.

Reviewer #3: (No Response)

PLOS authors have the option to publish the peer review history of their article (what does this mean?). If published, this will include your full peer review and any attached files.

Reviewer #1: No

Reviewer #2: No

Reviewer #3: No

Figure Files:

Data Requirements:

Reproducibility:

References

---

## [Decision Letter · Decision Letter 1]

18 Jun 2022

Dear Mrs Ledien,

We are pleased to inform you that your manuscript 'Linear and Machine Learning Modelling for Spatiotemporal Disease Predictions: Force-of-Infection of Chagas Disease' has been provisionally accepted for publication in PLOS Neglected Tropical Diseases.

Best regards,

Uwem Friday Ekpo, PhD

Associate Editor

Robert Reiner

Deputy Editor

Reviewer's Responses to Questions

**Key Review Criteria Required for Acceptance?**

**Methods**

-Are the objectives of the study clearly articulated with a clear testable hypothesis stated?

-Is the study design appropriate to address the stated objectives?

-Is the population clearly described and appropriate for the hypothesis being tested?

-Is the sample size sufficient to ensure adequate power to address the hypothesis being tested?

-Were correct statistical analysis used to support conclusions?

-Are there concerns about ethical or regulatory requirements being met?

Reviewer #1: Yes

**Results**

-Does the analysis presented match the analysis plan?

-Are the results clearly and completely presented?

-Are the figures (Tables, Images) of sufficient quality for clarity?

Reviewer #1: Yes

**Conclusions**

-Are the conclusions supported by the data presented?

-Are the limitations of analysis clearly described?

-Do the authors discuss how these data can be helpful to advance our understanding of the topic under study?

-Is public health relevance addressed?

Reviewer #1: Yes

**Editorial and Data Presentation Modifications?**

Reviewer #1: NA

**Summary and General Comments**

Reviewer #1: (No Response)

PLOS authors have the option to publish the peer review history of their article (what does this mean?). If published, this will include your full peer review and any attached files.

Reviewer #1: No

---

## [Editor Report · Acceptance letter]

15 Jul 2022

Dear Mrs Ledien,

We are delighted to inform you that your manuscript, "Linear and Machine Learning Modelling for Spatiotemporal Disease Predictions: Force-of-Infection of Chagas Disease," has been formally accepted for publication in PLOS Neglected Tropical Diseases.

Best regards,

Shaden Kamhawi

co-Editor-in-Chief

Paul Brindley

co-Editor-in-Chief
